# Visual Analogies between Atari Games for Studying Transfer Learning in RL

**Doron Sobol**[1], **Lior Wolf**[1,2] **& Yaniv Taigman**[2]
[1] School of Computer Science, Tel-Aviv University
[2] Facebook AI Research

## Abstract

In this work, we ask the following question: can visual analogies, learned in an unsupervised way, be used in order to transfer knowledge between pairs of games and even play one game using an agent trained for another game. We attempt to answer this research question by creating visual analogies between a pair of games: a source game and a target game. For example, given a video frame in the target game, we map it to an analogous state in the source game and then attempt to play using a trained policy learned for the source game. We demonstrate convincing visual mapping between four pairs of games (eight mappings). These mappings are used to evaluate three transfer learning approaches. The code and models are available at `https://github.com/doronsobol/Visual_analogies_for_RL_transfer_Learning`

## 1 Introduction

One of the most fascinating capabilities of humans is the ability to generalize between related but vastly different tasks. A surfer will be able to ride a snowboard after much less training than a beginner in board sports; a gamer experienced with adventure games will solve escape rooms long before the one hour is up; and a veteran tennis player will often top the office's ping pong league.

The goal of this work is to check if a Reinforcement Learning (RL) agent can gain such an ability: an actor is being trained and evaluated on a target task after learning a source task in the typical reinforcement learning setting. The actor is also provided with mappers that given a frame in either game, are able to generate the analogous frame in the other game.

The bidirectional mappers between the video sequences are based on recent approaches to the task of finding visual analogies, in combination with an added regularization term. We evaluate our methods on two groups of games, and are able to successfully learn the mappers between all same-group pairs.

Building on the existence of these mappers, we propose several Transfer Learning (TL) techniques for utilizing information from the source game when playing the target game. These methods include techniques such as data-transfer and distillation. Unfortunately, none of these methods seem to be consistently helpful, maybe with the one exception of the first, which uses scenes that are visually adapted from the source game to the domain of the target game.

Despite the moderate success, we believe that our work presents value to the community in multiple ways. First, we are successful at the challenging video conversion task, which could benefit future efforts. Second, we devise a few possible TL methods that "almost work". Third, a critical view of the practical value of TL in the current RL landscape is seldom heard. Lastly, by sharing our results, code, and models, we hope to help others in minimizing wasted efforts.

## 2 Settings and Methods

To create visual analogies between a pair of games, we collect frames off-line. The actor that is used to play the game does not need to be an expert and we do not imitate it. However, it is required that the states are diverse enough and, therefore, the actor is required to remain in the game for a while.

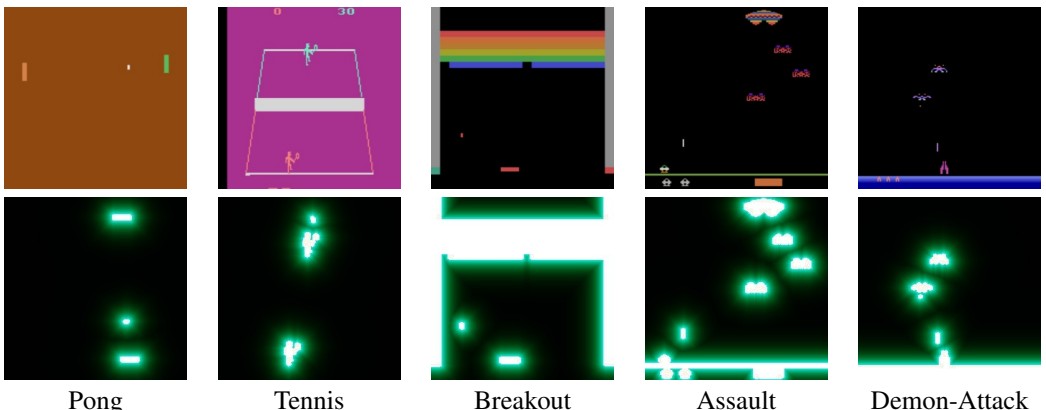

Figure 1: The obtained attention maps for a frame from each of the five tested games.

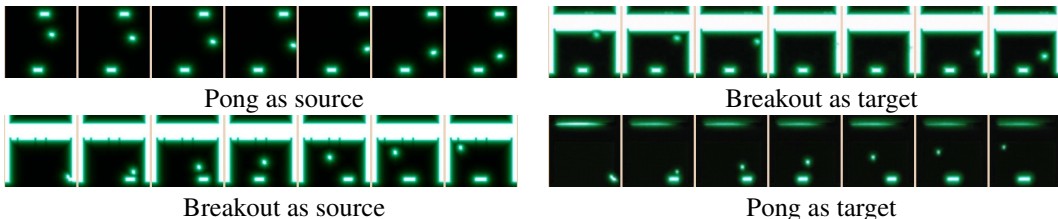

Figure 2: Images of samples of consecutive frames from the source game (left) to the target game (right). See Appendix A for the other games.

These frames are used to learn, in an unsupervised way, a mapper $G : s \to t$, between the frames of the source game $s$, and the frames of the target game $t$. We also learn the mapper in the reverse direction $G^{-1} : t \to s$.

We assume that we have the ability to train an agent in the source game without any limitation on the number of training episodes. Our goal is to be as efficient as possible in the training of the target game.

## 2.1 LEARNING CROSS-DOMAIN VIDEO MAPPING

The unsupervised learning step requires prior processing of the data. This includes the following steps: (a) Rotating the frames, if needed, so that the main axis of motion is horizontal. (b) Applying an attention operator to the frame, by subtracting either the median pixel value at each location or the median pixel value of the entire image (depending on the game), and then applying a threshold to obtain a binary image. (c) Applying a dilation filter on the image to enlarge the relevant objects. (d) Creating three channels by cloning the dilated image and applying two levels of blurring. The resulting images are shown in Fig 1.

To train the mapper functions $G$ and $G^{-1}$, we use the network architecture of UNIT GAN with cycle consistency loss (Liu et al., 2017). By itself, this method leads to mode collapse. In order to fix this, we add the gradient-penalty regularization term of improved WGAN (Gulrajani et al., 2017), adapted to the problem of cross-domain mapping:

$$L_{GP} = \mathbb{E}[(||\nabla_{\hat{x}} D(\hat{x})||^2 - 1)^2],$$

where $\hat{x}$ is either $\hat{s} = \epsilon s + (1 - \epsilon)G^{-1}(t)$ or $\hat{t} = \epsilon t + (1 - \epsilon)G(s)$, $D$ is the GAN's discriminator, and $\epsilon \sim U[0, 1]$.

## 2.2 TRANSFER LEARNING METHODS

Training the strategy $\pi_t$ for the target game and the strategy $\pi_s$ of the source game (when used), is done with the asynchronous actor-critic (A3C) algorithm (Mnih et al., 2016). The network architec-

Table 1: The level of success (see text) reached by the various methods.

| SOURCE ⇒ TARGET | DATA TRANSFER PRETRAINING | CONTINIOUS DATA TRANSFER | DISTILLATION |
|---|---|---|---|
| BREAKOUT ⇒ PONG | *,2 | - | - |
| PONG ⇒ BREAKOUT | *,2 | 2 | - |
| TENNIS ⇒ PONG | 3 | - | - |
| TENNIS ⇒ BREAKOUT | - | - | - |
| BREAKOUT ⇒ TENNIS | 1 | 1 | 1 |
| PONG ⇒ TENNIS | 1 | 1 | 1 |
| ASSAULT ⇒ DEMON-ATTACK | 1 OR 2 | - | - |
| DEMON-ATTACK ⇒ ASSAULT | 2 | - | 2 |

ture consists of four convolutional layers followed by an LSTM layer and two fully connect layers for the predicted action and value.

We tried various methods for transferring knowledge between games, including:

I **Data transfer for pretraining:** We transform frames from the source game $s$ using $G$. We train a policy $\pi_t$ on these frames using the reward of the source game and using a static mapping of actions, instead of the regular source game actions, and then fine tune the resulting policy on the target game.

II **Continuous data transfer:** Instead of pretraining $\pi_t$ on $G(s)$, we provide it with mixed samples from $G(s)$ and $t$ throughout the entire training process.

III **Distillation:** Directly fine-tunning $\pi_s$ failed, since it was trained on the source game. Fine-tunning $\pi_s \circ G^{-1}$ instead, lead to an overly complex network. We found it preferable to train a network of the same architecture as $\pi_t$ to mimic $\pi_s \circ G^{-1}$, on unsupervised frames from the target game. We then continue to train this network using real data.

## 3 RESULTS

The experiments are conducted on five Atari games, split into two groups. The first group contains the games Breakout, Tennis and Pong, in which the player has a paddle it controls and its goal is to hit the ball in order to achieve a certain objective. The second games are Demon-Attack and Assault, in which the player has a spaceship it controls and it needs to shoot the targets (similar to Space Invaders). We were not able to identify other potential pairs among the Atari games. The two groups give rise to four pairs of games, which yield eight transfer directions.

Samples of the transferred frames using the mapping method described in Sec. 2.1 can be found in Fig. 2, and in Appendix A. The mappings obtained seem to convey the semantics of the games.

We design a subjective rating scale for describing the level of success of the TL methods described in Sec. 2.2 on a given pair of games. A method is successful in transferring knowledge from the source game, if reaching a certain level of performance requires less supervised training samples than the baseline method of vanilla training in the target domain. We distinguish between three levels of success: (1) Upon convergence or reaching the maximum possible reward, the method that employs TL outperforms the baseline method. (2) The TL method achieves almost all levels of performance between the random performance and the converged performance with less samples than the vanilla method. (3) The TL method achieves non-trivial levels of performance faster than the baseline method but then stops leading. We also employ a star (*) to denote situations in which the TL method starts off, without any supervised samples from the target domain, in a level that is significantly better than random. This can happen with any level of success. Lastly, we employ a dash (-) to indicate the lack of success.

Tab. 1 shows the level of success reached by the various methods, in comparison to the baseline method. While the scoring is subjective, the table suggests that the data transfer for the purpose of pretraining is the only method to consistently outperform the baseline. Appendix B contains the full training logs.

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

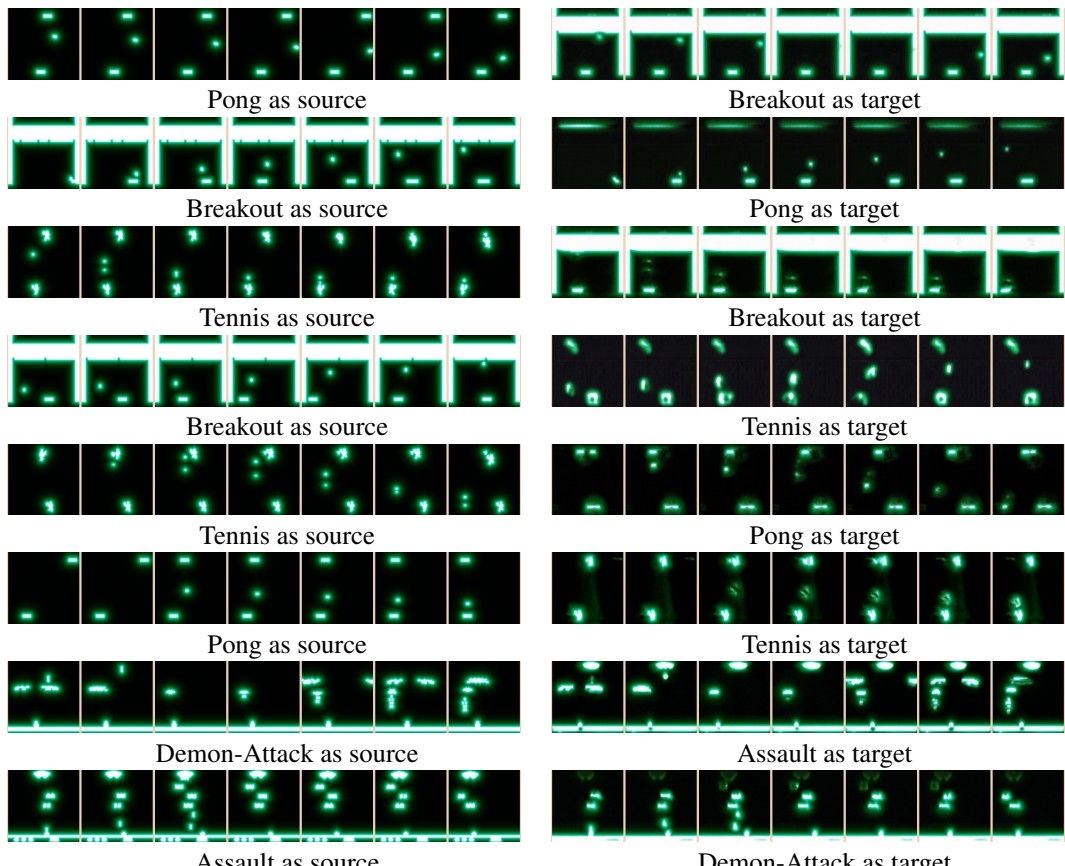

Figure 3: Images of samples of consecutive frames from the source game (left) to the target game (right).

## A  VISUAL TRANSFER

Fig.3 shows consecutive frames from the source games and their corresponding mapping in the target domain, using the trained function $G$.

## B  TRAINING PROGRESS PER GAME

Fig. 4 shows the training graphs of the transfer learning methods. Each point on the graphs is an average of samples of the model from the last 100K training states. The plotted results are the average of three independent runs.

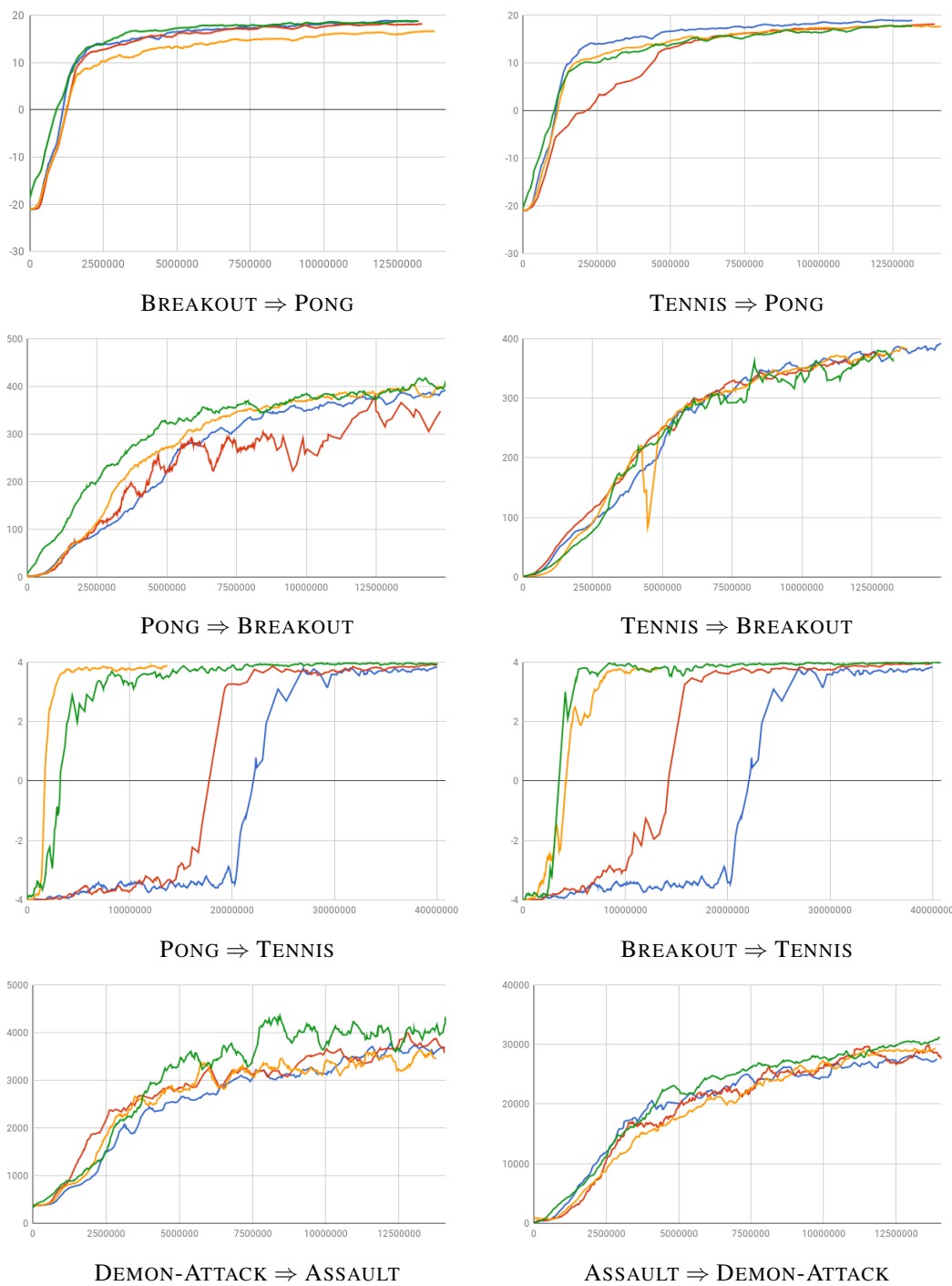

Figure 4: A comparison of the training logs for the various TL methods. The x-axis is the number of training steps, and the y-axis is the reward. The plots are averaged over three independent runs. The blue line is the baseline, the red is distillation, the yellow is continuous data transfer and the green is data transfer for pretraining.