# OpenReview forum: "Visual Analogies between Atari Games for Studying Transfer Learning in RL"
_ICLR.cc/2018/Workshop — Reject_

### Official Review · AnonReviewer2 · 2018-03-02
**A transfer learning technique lacking clear practical applications**

**Rating:** 4
**Confidence:** 3

**Review:**

This paper proposes an RL transfer learning algorithm between Atari games, based on mapping a game state representation (as pre-processed visual input) to another similar game’s state representation. This requires both games to have relatively similar state representations and game mechanics, for instance in both Pong and Breakout the player needs to catch a ball with a paddle at the bottom of the screen (after rotating Pong accordingly). Authors investigate three transfer learning schemes, with the best one consisting in pre-training the policy on game #1 with states mapped to game #2’s representation, before fine-tuning it on game #2.

This sounds like an intriguing idea, but to be honest I am not an expert in transfer learning, and the lack of related work makes it hard for me to assess its originality. Although I am not very familiar with this research, I know there is active work on transferring in particular policies learned in simulators into the real world, and it seems to me that there should be links to be made with such approaches. In addition, the (GAN-based) technique to map states from one game to another is presented way too briefly for me to understand how it works (and in particular how much data from the target game is required, since this can be a constraint in a transfer learning scenario). Images from Fig. 2 show that it learned a rather « naive » mapping, with the game-specific elements (at the top of the screen) being generated essentially as constant pixels. This is not particularly surprising since there is no obvious true answer to this mapping task, but it suggests it is going to be hard to learn meaningful policies from such representations, where important state information may not be available (I can see how it can still work to some extent here because for instance just being able to catch the ball is already a decent strategy in both Pong and Breakout, but this may not always be the case). Overall the usefulness of this method seems limited to me, and there is no discussion on potential practical applications (nor on the reasons why / when the proposed techiques may or may not work).

On the positive side, I appreciate that the authors are willing to share some negative results as well, along with the associated code.

I would also like to suggest another related transfer learning strategy that might be worth experimenting with: train a policy pi_s on frames G^-1(G(source game frame)), then pre-train pi_t to mimick pi_s(G^-1(target game frame)). This is close to the « Distillation » strategy but with the difference that pi_s would be trained on mapped images, which might work better (?)

---

### Official Review · AnonReviewer1 · 2018-03-05
**Visual Analogies may not be enough for transfer learning between Atari games**

**Rating:** 4
**Confidence:** 4

**Review:**

Summary: Visual analogies between individual frames in pairs of (source & target) Atari  games are employed in the service of policy learning on the target game. Several transfer methods are used in order to take advantage of visual analogies, but results are mixed with the exception of transfer to the game of Tennis, where data efficiency seems better.

Pros:
Fairly generic video conversion methods were used, based on GANs.
Several knowledge transfer methods were evaluated, including dataset augmentation, distillation and pre-training.
In my mind, the paper provides convincing evidence suggesting that visual analogies are *not* sufficient in the Atari domain.

Cons:
Domain knowledge in the form of game-specific image pre-processing is used to facilitate the process.
Some amount of manual data collection is used, which makes it difficult to infer how the method would perform with a different human in the loop or with a different game were expert trajectories are not available.
It is very unlikely that states in Atari video-games would be generally well characterized by single frames, especially in fast paced games such as Breakout. This is because it is not possible to infer a ball’s direction from a single frame. I suggest using frame sequences as the starting point for state analogies.

Notes/Questions:
Perhaps the authors could clarify why visual analogies are supposed to be sufficient for transfer. IMHO it does not follow that similar frames should lead to similar actions, not even within a single game. In fact, state aliasing is a well known problem in (deep) reinforcement learning [1].
Considering only visual analogies ignores games dynamics, which is obviously dangerous in fast-paced, ”reflex” based games as those considered; for example, ignoring dynamics implicitly assumes the ball is moving at the same speed in both games, which is obviously not true, since the ball accelerates across the game of Breakout.
It may be advisable to evaluate the components of the approach in isolation. Would you see transfer if you were to use Vertically Flipped Pong as your source game and transfer to regular Pong? What about Horizontally Flipped Pong as a source. I suggest perfecting the method such that these ‘artificial’ transfer problems work well.
Since you’re using GANs, are all the hand-engineer, game-specific transformations really required? They severely restrict the applicability of the method, imho.
There may exist domains where visual analogies are indeed sufficient, but different Atari games may not be the best examples, particularly due to different dynamics of every game.


[1] E.g. https://arxiv.org/abs/1707.06887

---

### Official Review · AnonReviewer3 · 2018-03-12

**Rating:** 5
**Confidence:** 4

**Review:**

The authors propose a brief study on the effectiveness of certain kinds of transfer-learning in RL tasks, employing visual-analogies as the basis for transferring information.

The paper is reasonably straightforward to follow -- although there are a couple of places where the exposition is short of detail and difficult to follow (e.g. Section 2.2).

Some issues I encountered:
1. The authors claim ".. we are successful at the challenging video conversion task ..."
   This is somewhat difficult to substantiate for two reasons:
   a. There is a fairly substantial amount of manual feature-extraction/augmentation to enable this conversion, and
   b. The is no quantitative evaluation of it -- just a few qualitative examples with not too much in the way of explanation of what to look for.

2. They also state "... a critical view of the practical value of TL in the current RL landscape is seldom heard."
   The manuscript does not state anything about the value of TL for RL. If the authors intended the lack of success they faced as evidence for it, I'm afraid that doesn't quite fit. If anything, the lack of success points to the difficultly in making TL work in RL, not so much about the value of doing TL in/for RL.

3. The description of transfer-learning methods in Section 2.2 is somewhat difficult to follow; particularly the 'distillation' part. It's not clear what the authors refer to as "fine-tuning" here.
   When they say "Directly fine tunning[sic] $\pi_s$ failed ...", do they mean they were trying to directly adapt a policy learnt on the source game in the target game?

4. It's not clear if the different 'levels of success' proposed are actually computed off-of significant values. In the graphs reported in the appendix, the authors state that the mean of three runs was plotted, but the std-deviation/variance is not present.
Without this, it's really difficult to say if, say the Breakout->Tennis transfer, actually does outperform the baseline (the difference in asymptotic reward obtained is almost negligible), or if the ordering is simply what it is because of randomness.

Overall, I feel that the manuscript could have more value with a careful analysis of results [the significance (rather, the potential lack of) of results worries me] and a cleaned-up exposition.

---

### Decision · Program_Chairs · 2018-03-20
**ICLR 2018 Workshop Acceptance Decision**

**Decision:**

Reject

**Comment:**

Based on the reviews, this paper has not been accepted for presentation at the ICLR workshop. However, the conversation and updates can continue to appear here on OpenReview.